# Distilling Motion Planner Augmented Policies into Visual Control Policies for Robot Manipulation

**I-Chun Arthur Liu**[2*]  **Shagun Uppal**[1*]  **Gaurav S. Sukhatme**[2†]  **Joseph J. Lim**[1‡]
**Peter Englert**[2]  **Youngwoon Lee**[1§]
[1]Cognitive Learning for Vision and Robotics Lab      [2]Robotic Embedded Systems Laboratory
University of Southern California

**Abstract:** Learning complex manipulation tasks in realistic, obstructed environments is a challenging problem due to hard exploration in the presence of obstacles and high-dimensional visual observations. Prior work tackles the exploration problem by integrating motion planning and reinforcement learning. However, the motion planner augmented policy requires access to state information, which is often not available in the real-world settings. To this end, we propose to distill a state-based motion planner augmented policy to a visual control policy via (1) visual behavioral cloning to remove the motion planner dependency along with its jittery motion, and (2) vision-based reinforcement learning with the guidance of the smoothed trajectories from the behavioral cloning agent. We evaluate our method on three manipulation tasks in obstructed environments and compare it against various reinforcement learning and imitation learning baselines. The results demonstrate that our framework is highly sample-efficient and outperforms the state-of-the-art algorithms. Moreover, coupled with domain randomization, our policy is capable of zero-shot transfer to unseen environment settings with distractors. Code and videos are available at `https://clvrai.com/mopa-pd`.

**Keywords:** Visual Policy Distillation, Motion Planning, Reinforcement Learning

## 1   Introduction

Solving complex manipulation tasks in obstructed environments is a challenging problem in deep reinforcement learning (RL) since it requires precise object interactions as well as collision-free movement across obstacles. To tackle this problem, prior works [1–3] have proposed to combine the strengths of motion planning (MP) and RL – safe collision-free maneuvers of MP and sophisticated contact-rich interactions of RL, demonstrating promising results. However, MP requires access to the geometric state of an environment for collision checking, which is often not available in the real world, and is also computationally expensive for a real-time control. To deploy such agents in realistic settings, we need to resolve the dependency on the state information and costly computation of MP, such that the agent can perform a task in the visual domain.

To this end, we propose a two-step distillation framework, motion planner augmented policy distillation (MoPA-PD), that transfers the state-based motion planner augmented RL policy (MoPA-RL [1]) into a visual control policy, thereby removing the motion planner and the dependency on the state information. Concretely, our framework consists of two stages: (1) visual behavioral cloning (BC [4]) with trajectories collected using the MoPA-RL policy and (2) vision-based RL training with the guidance of smoothed trajectories from the BC policy. The first step, visual BC, removes the dependency on the motion planner and the resulting visual BC policy generates smoother behaviors compared to the motion planner's jittery behaviors. Then, in the second step, we further improve the visual policy

---

*Equal contribution

†Gaurav Sukhatme holds concurrent appointments as a Professor at USC and as an Amazon Scholar. This paper describes work performed at USC and is not associated with Amazon.

‡AI Advisor at NAVER AI Lab

§Work done during an internship at NAVER AI Lab

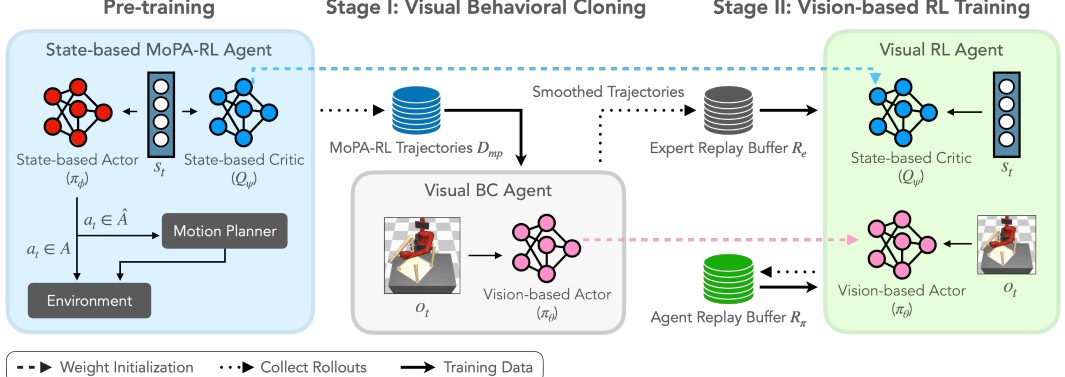

Figure 1: Our two-step framework distills the state-based MoPA-RL agent into a visual control policy. In stage 1, we learn a visual BC policy using MoPA-RL trajectories, thereby ensuring smooth trajectories for the expert replay buffer $\mathcal{R}_e$, while removing the motion planner. In stage 2, we learn a BC trajectory-guided vision-based RL agent with an asymmetric actor-critic algorithm using both the expert replay buffer $\mathcal{R}_e$ and agent replay buffer $\mathcal{R}_\pi$. To improve sample efficiency, we initialize the actor and the critic of the vision-based agent with the weights of the BC policy and the MoPA-RL critic, respectively.

using asymmetric actor-critic RL [5] directly from the image observations to overcome sub-optimality in the MoPA-RL policy through self-exploration.

For efficient vision-based RL training, our method leverages the experience of the BC policy and the state-based critic from MoPA-RL by initializing the critic network with the MoPA-RL critic and the actor network through BC pre-training; BC-trajectory guided RL training using a separate expert replay buffer; tuning the entropy coefficient to encourage exploitation (i.e., maximizing reward) over exploration (i.e., maximizing entropy).

In summary, our contributions are as follows:

- We propose a novel framework to learn a visual control policy in cluttered scenarios by distilling a state-space policy that uses a motion planner into an image-space policy by removing the motion planner dependency.
- Our asymmetric visual policy leaning ensures high sample-efficiency using weight initialization followed by BC trajectory-guided RL with entropy coefficient tuning.
- The distilled visual policy is capable of zero-shot domain transfer to unseen environments with visual domain randomization during training.
- Our method outperforms the state-of-the-art methods in terms of success rate, sample efficiency, and path length on three manipulation tasks in obstructed environments.

## 2 Related Work

The objective of our work is to learn complex manipulation tasks in realistic and obstructed environments. Motion planner augmented RL methods [1–3] have shown promising results in solving complex tasks in obstructed environments by combining MP and RL. However, these methods cannot be deployed in real-world settings due to their dependency on state information. To remove this dependency, state-based policies must be transferred to the visual domain.

A typical approach to distill state-based policies into vision-based policies is behavioral cloning (BC [4]). However, BC often suffers when encountering states unseen during training [6]. Therefore, prior works have proposed various methods to improve BC agents using offline RL [7] and online RL [8]. However, these works do not involve the distillation of motion planners.

Several recent efforts have been made towards distilling MP algorithms into neural motion planners using learning-based methods [9–13]. Most of these efforts can be broadly categorized into imitation

learning (IL) and RL paradigms. For IL, supervised learning on MP trajectories has been used to learn neural network-based planners [10–14]. However, the performance of such supervised learning approaches are limited by the collected dataset and demonstrator's performance.

To discover better policies with additional interactions, various off-policy RL methods have been studied for superseding motion planners with neural network policies [13, 15, 16]. These approaches utilize expert trajectories stored in the replay buffer for guided exploration, which leads to better policies than experts [17–20]. However, these works learn either in unobstructed environments [17–20] or on simpler tasks not involving object manipulation [13]. They also assume fully observable environments and learn policies in state space. In this work, we focus on complex manipulation tasks in obstructed environments using visual observations.

Moreover, most offline RL approaches [21, 22] in sparse reward scenarios are coupled with hindsight experience replay (HER [23]). They empirically show that not using HER significantly hurts the agent's performance [13, 5, 16, 17]. However, HER is designed to efficiently train goal-conditioned policies for multi-goal RL, which makes it unsuitable for tasks with multiple sequential steps without explicitly conditioning on goals. In contrast, our method successfully learns composite manipulation tasks in cluttered environments without using HER for tackling sparse rewards.

# 3   Method

Our goal is to solve complex manipulation tasks in obstructed environments with visual inputs. While MP-based methods can efficiently solve such tasks, they are restricted to learning a state-based policy which is difficult to transfer to real environments. Moreover, sampling-based motion planning incurs significant computational costs, making it hard to integrate them into real-time controllers. Thus, we propose a method that distills an MP-augmented state-based agent into a visual policy, removing the MP and state dependency. We formally define our problem and introduce the MP-augmented RL in Section 3.1. Then, we describe our two-step visual distillation approach in Section 3.2.

## 3.1   Preliminaries

**Problem formulation**   We formulate our problem as a Markov Decision Process (MDP) $M$ defined by a tuple $(\mathcal{S}, \mathcal{O}, \mathcal{A}, P, R, \rho_0, \gamma)$ that consists of a state space $\mathcal{S}$, partial observation space $\mathcal{O}$ (visual inputs corresponding to states), action space $\mathcal{A}$, transition function $P : \mathcal{S} \times \mathcal{A} \times \mathcal{S} \to \mathbb{R}$, reward function $R$, initial state distribution $\rho_0$, and discount factor $\gamma$. The agent is represented as a policy $\pi(a_t|o_t)$, which takes an action $a_t$ under a visual observation $o_t$ and receives a reward $r_t = R(s_t, a_t)$, with the state transitioning to $s_{t+1}$. The goal of the agent is to maximize the expected discounted sum of rewards $\mathbb{E}_{(s,a)\sim\pi} \sum_{t=0}^{T-1} \gamma^t r_t$, where $T$ is the episode horizon.

**Motion Planner-Augmented RL (MoPA-RL)**   To tackle manipulation tasks in cluttered environments, Yamada et al. [1] proposed to incorporate MP and RL via an MP-augmented MDP $\tilde{M} = (\mathcal{S}, \tilde{\mathcal{A}}, \tilde{P}, \tilde{R}, \rho_0, \gamma)$ that consists of the state space $\mathcal{S}$, enlarged action space $\tilde{\mathcal{A}}$ augmenting the direct action space $\mathcal{A}$ with the MP action space $\hat{\mathcal{A}}$,[5] augmented transition function $\tilde{P} : \mathcal{S} \times \tilde{\mathcal{A}} \times \mathcal{S} \to \mathbb{R}$, augmented reward function $\tilde{R}(s, \tilde{a})$, initial state distribution $\rho_0$, and discount factor $\gamma$.

MoPA-RL [1] learns a policy $\pi_\phi(\tilde{a}_t|s_t)$ on the augmented MDP $\tilde{M}$ with an off-policy RL algorithm, SAC [24]. Precisely, given a state $s_t$, the policy predicts an action $\tilde{a}_t$, which is defined as a robot joint displacement $\Delta q_t$. If the action lies within the direct action space $\mathcal{A}$, it is directly executed by the controller as the agent performs sophisticated and contact-rich manipulations. However, when the actions are larger in magnitude (i.e. $\tilde{a} \in \hat{\mathcal{A}}$), the probability of collisions in the presence of obstacles increases. Therefore, a sampling-based motion planner, RRT-Connect [25], is called to realize such actions with large joint displacements by computing collision-free paths. This allows the agent to efficiently explore obstructed environments while avoiding collision [1].

---

[5]MoPA-RL [1] defines the direct action space as $\mathcal{A} = [-\Delta q_{\text{step}}, \Delta q_{\text{step}}]^d$, where $\Delta q_{\text{step}}$ is the maximum joint displacement, and the enlarged MP-augmented action space as $\tilde{\mathcal{A}} = [-\Delta q_{\text{MP}}, \Delta q_{\text{MP}}]^d$, where $\Delta q_{\text{MP}}$ is the motion planner action limit with $\Delta q_{\text{MP}} > \Delta q_{\text{step}}$. In MoPA-RL, actions in the direct action space $a \in \mathcal{A}$ are directly applied as a joint torque while other large actions $\hat{a} \in \hat{\mathcal{A}} = \tilde{\mathcal{A}} \setminus \mathcal{A}$ invoke motion planning. We refer the readers to Yamada et al. [1] for more details.

### 3.2 Motion Planner Augmented Policy Distillation

Despite the advantages of MoPA-RL, its applicability to the real world is limited due to the high computational cost of MP and dependency on fully observable environment states. To this end, we propose a visual distillation method, motion planner augmented policy distillation (MoPA-PD), that learns an image-based control policy (without MP) from a state-based MP-augmented policy by leveraging its rollouts and the learned state-based critic as a guidance. Concretely, our proposed method consists of two stages, as illustrated in Figure 1. Given a state-based MoPA-RL agent, we first use BC to train a vision-based actor with trajectories collected from the MoPA-RL policy (Section 3.2.1), and then we further improve the vision-based agent via BC trajectory-guided asymmetric actor-critic RL [5] with the visual BC actor and MoPA-RL critic (Section 3.2.2).

#### 3.2.1 Stage 1: Visual Behavioral Cloning and Trajectory Smoothing

The MoPA-RL policy is learned with access to complete information about the environment states and also utilizes MP for executing large action steps without collisions. In this paper, we aim to utilize demonstrations from this planner-based policy and distill it into a visual control policy, thereby deducting expensive MP computations and training the actor in visual domain. With the learned MoPA-RL policy, we first collect multiple transitions $d_i$ in low-level (direct) action space and store them into the MoPA-RL demonstration dataset $\mathcal{D}_{\mathrm{mp}} = \{d_1, d_2, d_3, ...\}$, where $d_i = (s_i, o_i, a_i, r_i, s_{i+1})$. Then, we train a visual BC actor $\pi_\theta(a_t|o_t)$ using observation-action pairs $(o_i, a_i)$ from the dataset $\mathcal{D}_{\mathrm{mp}}$ by minimizing the mean squared error.

Distilling the MoPA-RL trajectories using BC not only enables the BC actor to work directly on visual inputs but also reduces jerky motion planning behaviors of MoPA-RL, which occur due to motion planner's priority on obstacle avoidance over trajectory smoothness [26]. By removing unnecessary, jerky movements through visual BC, the resulting trajectories become smoother and even shorter, and thus help learning consistent and smooth motions.

However, the BC actor often fails when it encounters states not seen during training due to covariate shift [6]. To improve the robustness of the policy, we further train the visual BC actor using RL with additional environment interactions. In other words, the BC actor can be a good starting policy for visual RL training and the trajectories collected from the BC actor can effectively guide exploration.

#### 3.2.2 Stage 2: Vision-based RL with Asymmetric Actor-Critic

In the second stage, we further train the visual policy using RL directly on image observations to enhance the robustness of the policy and overcome sub-optimality in the planner-based policy. For efficient RL training, we adopt an asymmetric actor-critic architecture [5], where the actor acts based on environment images and robot joint states while the critic learns from environment states. This architecture is motivated by learning transferable visual policies, which do not require state information during inference but benefit from them while learning in simulation. Our training procedure comprises of the following components:

**Weight initialization of actor and critic networks**  Initializing actor and critic networks with suitable weights can accelerate RL training by providing more optimal rollouts from the actor and informative learning signals from the critic compared to the randomly initialized actor and critic [27, 28]. Especially when learning from pixels, suitable initialization can significantly improve sample efficiency [20, 29]. To this end, we propose to leverage the state-based agent's experience by utilizing its weights for initializing our asymmetric visual agent's networks, thereby bringing their initial distributions closer. Thus, the visual actor network $\pi_\theta(a_t|o_t)$ is initialized with the weights of the visual BC actor learned in Section 3.2.1. We also initialize the critic $Q_\psi(s_t, a_t)$ with the critic of MoPA-RL. Note that the action spaces of two critic networks are different, but initialization is still feasible since the action space of MoPA-RL is a superset of the direct action space.

**BC trajectory-guided asymmetric RL**  After initializing the actor and critic networks, we collect agent rollouts into the agent replay buffer $\mathcal{R}_\pi$. As an RL algorithm, we adopt Soft Actor-Critic (SAC [24]), an off-policy model-free RL algorithm for continuous control. We optimize our asym-

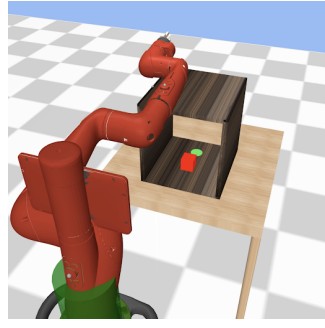 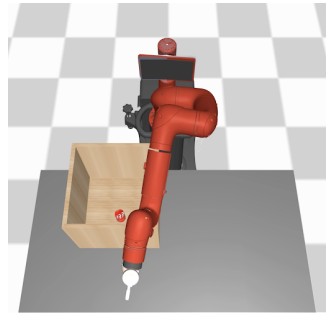 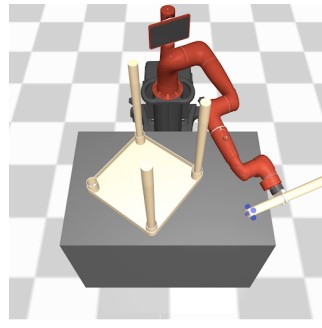

(a) Sawyer Push       (b) Sawyer Lift       (c) Sawyer Assembly

Figure 2: Manipulation tasks in obstructed environments from Yamada et al. [1]. (a) Sawyer Push: Push the red cube in the box to the green goal. (b) Sawyer Lift: Lift out the can inside the box. (c) Sawyer Assembly: Insert the table leg in the hole in the table top.

metric visual policy $\pi_\theta$ by maximizing the following objective:

$$J(\pi_\theta) = \sum_{t=0}^{T} \mathbb{E}_{(\mathbf{s_t}, \mathbf{a_t}) \sim \rho_{\pi_\theta}} \left[ Q_\psi \left( \mathbf{s}_t, \mathbf{a}_t \right) + \alpha \mathcal{H} \left( \pi_\theta \left( \cdot \mid \mathbf{o}_t \right) \right) \right], \tag{1}$$

where the temperature parameter $\alpha$ balances between exploration and exploitation using entropy $\mathcal{H}$.

RL training with a BC-initialized policy often quickly deviates from the original policy. To prevent this problem and guide exploration during RL, we update our RL agent not only with agent trajectories but also with BC policy (i.e. expert) trajectories [18]. Thus, we collect smoothed trajectories from our visual BC policy, $(s_i, o_i, a_i, r_i, s_{i+1})$, in an expert replay buffer $\mathcal{R}_e$. Then, for each RL training iteration, we sample 1:3 transitions from $\mathcal{R}_e$ and $\mathcal{R}_\pi$, respectively. A separate expert replay buffer ensures guided exploration from the expert and also circumvents catastrophic forgetting after weight initialization. Note that we use the BC trajectories $\mathcal{R}_e$ instead of the MoPA-RL data $\mathcal{D}_{\text{mp}}$ because jittery motion planning paths in the MoPA-RL data make RL training difficult and sub-optimal.

**Entropy coefficient tuning**    The performance of SAC is known to be sensitive to $\alpha$ [30]. Since our asymmetric agent already starts with the well-trained BC actor and MoPA-RL critic, we initialize $\alpha$ to values lower than the final $\alpha$ obtained in MoPA-RL. This is to ensure that with prior knowledge in the state-based agent, the visual agent focuses more on maximizing rewards over exploration. We describe the hyperparameter choice and implementation details in Section A.1 and Section B.

In summary, we propose a two-step visual distillation method for a state-based MoPA-RL agent using visual BC followed by BC trajectory-guided vision-based RL to remove the dependency on MP and environment states. For efficient RL training, our method leverages weight initialization of the actor and critic, the expert replay buffer of BC smoothed trajectories, and entropy coefficient tuning.

## 4 Experiments

In this paper, we propose to distill a motion planner augmented policy into a visual control policy for complex manipulation tasks in obstructed environments. In our experiments, we aim to answer the following questions: (1) Does IL efficiently learn policies for obstructed environments in image space? Moreover, is naively combining IL with RL sufficient for solving complex tasks? (2) Is distillation better than directly learning a visual policy using MP? (3) Does our approach, MoPA-PD, efficiently learn a visual policy using prior state-based exploration knowledge? (4) Is our visual policy capable of domain transfer and robust to unseen distractors?

### 4.1 Environments

We evaluate our approach on three obstructed environments (see Figure 2) from Yamada et al. [1], simulated using the MuJoCo physics engine [31]. We use a 32x32 image as a visual observation.

| | Sawyer Push | | | Sawyer Lift | | | Sawyer Assembly | | |
|---|---|---|---|---|---|---|---|---|---|
| | ASR ↑ | AEL ↓ | ADR ↑ | ASR ↑ | AEL ↓ | ADR ↑ | ASR ↑ | AEL ↓ | ADR ↑ |
| MoPA-RL | 98.2 | 111.0 | 52.7 | 95.0 | 109.2 | 52.7 | 99.8 | 63.3 | 82.4 |
| BC-Visual | 99.4 | 118.0 | 46.9 | 62.0 | 108.8 | 34.6 | 97.0 | 115.1 | 50.4 |
| Asym. SAC | 0.0 | 250.0 | 0.0 | 0.0 | 250.0 | 0.2 | 0.0 | 250.0 | 3.4 |
| MoPA-Asym. SAC | 0.0 | 250.0 | 0.0 | 0.0 | 250.0 | 0.0 | 0.0 | 250.0 | 0.0 |
| CoL | 0.0 | 250.0 | 0.0 | 29.8 | 173.3 | 9.7 | 0.0 | 250.0 | 0.0 |
| CoL w/ BC smoothing | 0.0 | 250.0 | 0.0 | 0.0 | 250.0 | 5.1 | 0.0 | 250.0 | 0.0 |
| Ours w/o BC smoothing | **100.0** | 34.0 | 108.7 | **99.4** | 43.6 | 100.7 | 0.0 | 250.0 | 0.0 |
| Ours | **100.0** | **32.0** | **110.8** | 99.0 | **42.0** | **101.7** | **100.0** | **61.7** | **84.5** |

Table 1: Average success rate (ASR), episode length (AEL), and discounted return (ADR) of our method and baselines averaged over five seeds. Each method is evaluated after 3M environment steps. MoPA-RL [1] is the expert agent trained with state-based policy. All baselines below horizontal line are trained with asymmetric actor-critic for fair comparison. Maximum horizon is 250 for all tasks.

- **Sawyer Push:** A Rethink Sawyer robot arm with 7 DoF, initially located outside the box, needs to reach an object placed inside the box and push it to the goal location within the box.
- **Sawyer Lift:** The Sawyer arm must reach an object placed inside a box, grasp, and lift it outside the box, avoiding collisions. The Sawyer arm is always initialized outside the box.
- **Sawyer Assembly:** The Sawyer arm needs to assemble the fourth leg (already attached to the gripper) of a table into the vacant hole while avoiding collisions to other three legs. This environment is built upon the IKEA furniture assembly environment [32].

For Sawyer Push and Sawyer Assembly, the agent receives a reward proportional to the distance between the end-effector and object, or the object and goal position when the distance is less than $\epsilon$. We use $\epsilon = 0.1$ for Sawyer Push and $\epsilon = 0.3$ for Sawyer Assembly while the distance between initial and goal state is around 1.2. An episode is considered successful when the distance between the cube and goal (Sawyer Push) or the peg-head and goal (Sawyer Assembly) is less than 0.05. For Sawyer Lift, we use the reward function similar to Fan et al. [33], where a reward is defined for all three intermediate stages of the task: *reach*, *grasp* and *lift*. A bonus reward signal of 150 is received upon successful task completion in all the environments. We refer readers to Section D for more details.

## 4.2 Baselines

We compare our method with the following baselines to evaluate the merits of our approach:

- **MoPA-RL** [1]: A MP-augmented policy, for learning large displacement actions with MP and smaller actions with RL. This policy also serves as our state-based expert agent.
- **Visual Behavioral Cloning (BC-Visual)**: A behavioral cloning policy trained on the image-action pairs collected from MoPA-RL trajectories.
- **Asym. SAC** [5]: An asymmetric actor-critic method, trained using SAC where the critic is learned in the state space and the actor is learned in the image space.
- **MoPA Asym. SAC**: MoPA-RL policy [1] learned using the asymmetric framework. Note that this method still uses a motion planner with an augmented action space. This method is a direct attempt at learning a visual policy using a MP-augmented framework [1].
- **CoL** [29]: A policy learned using the Cycle-of-Learning framework, which is the state-of-the-art algorithm for learning from demonstrations (LfD).
- **CoL (w BC Smoothing)** [29]: A policy similar to CoL [29], with BC for trajectory smoothing.
- **Ours (w/o BC Smoothing)**: A policy learned using our approach described in Section 4 without BC trajectory smoothing, i.e., we directly use MoPA-RL trajectories in $\mathcal{R}_e$.

## 4.3 Evaluation

We compare our approach against baselines on the following evaluation metrics averaged over five random seeds and 100 unseen episodes per seed:

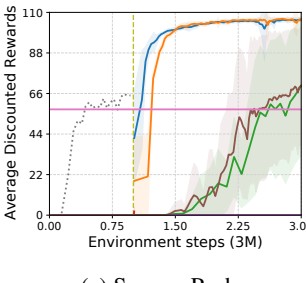 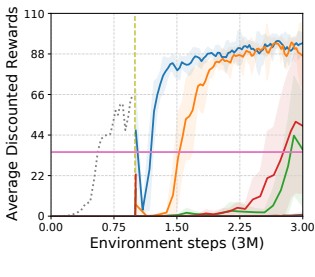 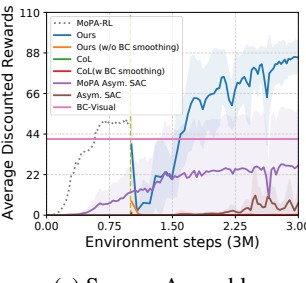

|            (a) Sawyer Push           |            (b) Sawyer Lift           |          (c) Sawyer Assembly          |

Figure 3: Learning curves of our method compared to baselines. All methods are trained for 3M environment steps. For the methods that require MoPA-RL, we train MoPA-RL for 1M steps and then train the methods for 2M steps (total 3M) for a fair comparison. Our method solves all the tasks with the highest average discounted return. Our method w/o BC smoothing can solve two tasks with slower convergence compared to Ours, but fails on Sawyer Assembly.

- **Average Success Rate (ASR)** is the average number of successful episodes.
- **Average Episode Length (AEL)** is the average length of successful episodes.
- **Average Discounted Return (ADR)** is the average discounted sum of rewards $\sum_{t=0}^{T-1} \gamma^t R(s_t, a_t)$, with $T$ being the episode horizon. An episode completed in more time steps has a lower discounted reward, due to exponentially discounted reward with $\gamma = 0.99$.

## 4.4 Results

**Comparisons with baselines**    Figure 3 illustrates the learning curves of our method and all other baselines using discounted rewards against the number of training steps. As per the trend, our method is far more sample-efficient and outperforms all the baselines. Asym. SAC [5] fails to directly learn a visual policy for Sawyer Lift and Sawyer Assembly in obstructed environments, where exploration is hard. Moreover, MoPA Asym. SAC, which is a direct attempt at learning a visual policy while using MP, does not successfully learn to solve the tasks. CoL [29] which has a straightforward combination of RL and BC objective for actor optimization receives partial rewards for some tasks at around 1.5-2M steps, much slower than our method's convergence.

As reported in Table 1, BC achieves decent performance for Sawyer Push and Assembly, but not for the Sawyer Lift task. However, it achieves high AEL and low ADR for all tasks, showing its inefficiency to solve the task fast. This is because BC is bound to perform as good as the expert demonstrations (here from MoPA-RL) at its best. In contrast, our method optimizes trajectories beyond expert signals and achieves significantly lower AEL and higher ADR compared to baselines.

**Ablation on BC trajectory smoothing**    Our method without BC smoothing has higher variance across seeds and converges slower than our method for two tasks (see Figure 3). For Sawyer Assembly, it does not learn to solve the task at all. In short, BC smoothing of the motion planner trajectories is important for our method to work across all environments. This is because the motion planner based trajectories are usually jittery and non-smooth. Using behavioral cloning refines each transition, thereby making the entire trajectories much smoother and help in efficiently training the RL agent.

**Ablation on weight initialization**    We compare our method with and without actor-critic initialization in appendix, Figure 7 and do not observe any episode success until 1.2M environment steps for the latter. This shows the importance of our proposed weight initialization for learning in the visual domain. We further elaborate the ablation results in Section A.2.

**Ablation on entropy coefficient tuning**    We examine the effect of different values of $\alpha$ on the trade-off between entropy maximization and reward maximization for the SAC objective as shown in Section A.1. Compared to higher $\alpha$ values, smaller values of $\alpha$ improve the sample efficiency during visual policy learning. Since we utilize prior knowledge from the state-based agent, we use a smaller alpha to exploit the previously acquired knowledge instead of exploring the entire state space again.

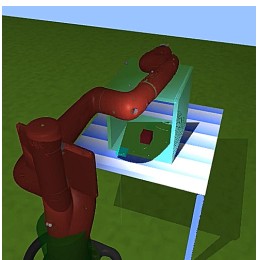 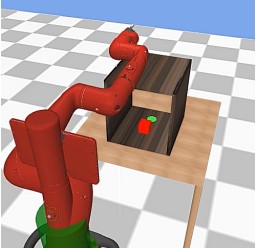 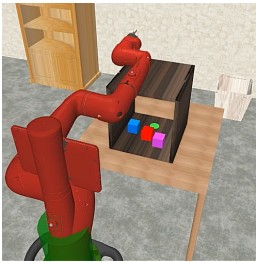 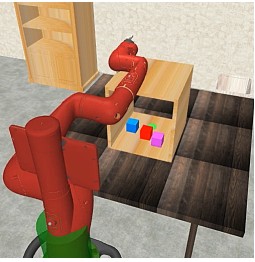

(a) Domain randomization    (b) Original environment    (c) Scenario 1    (d) Scenario 2

Figure 4: Illustration of environments for domain transfer. (a) Domain randomized environments are used for training; For testing, we use (b) original environment without randomization, (c) Scenario 1 with distractors such as unseen cubes (blue and magenta), walls, floor, and furniture, and (d) Scenario 2 similar to Scenario 1 along with variation in size of the table and texture of the box and table.

| | Sawyer Push | | | Sawyer Lift | | | Sawyer Assembly | | |
| --- | --- | --- | --- | --- | --- | --- | --- | --- | --- |
| | ASR ↑ | AEL ↓ | ADR ↑ | ASR ↑ | AEL ↓ | ADR ↑ | ASR ↑ | AEL ↓ | ADR ↑ |
| Original env | 99.7 | 39.1 | 103.1 | 99.3 | 37.3 | 106.5 | 100.0 | 49.7 | 93.4 |
| Scenario 1 | 100.0 | 40.4 | 102.2 | 96.7 | 38.3 | 103 | 100.0 | 47.9 | 94.8 |
| Scenario 2 | 99.3 | 43.4 | 99.04 | 97.3 | 37.0 | 104.7 | 100.0 | 47.7 | 95.0 |

Table 2: Evaluation metrics for domain transfer to unseen environments, illustrated in Figure 4.

## 4.5   Policy Transfer to Different Domains

In this experiment, we learn our policy with domain randomization (DR) to verify its domain transfer capabilities. DR is a promising approach for modelling transferable policies using a multitude of variations in simulation during training [34, 35], which makes the policy invariant to changes that are insignificant for the task. For this work, we randomize the simulation environment using different textures, colors, and lighting conditions for training as shown in Figure 4 and Figure 10. We then test our DR policy on three unseen scenarios, as illustrated in Figure 4 comprising of (1) the original environment without any randomization; (2) Scenario 1 with realistic background distractions (e.g., furniture, walls, and floor) and unseen distractor cubes (blue and magenta) near the target cube (red); and (3) Scenario 2 with additional changes in the texture of the table and box, and different size of the table. Our policy attains more than 96% ASR in all three unseen scenarios for all the tasks (see Table 2). This illustrates the robustness of our proposed framework in learning transferable visual policies, robust to unseen distractors in a sample-efficient manner.

## 5   Conclusion

In this paper, we introduce a two-step distillation method for learning manipulation tasks in obstructed environments in the visual domain. In step one, we use a MP-augmented RL policy as the state-based expert and subsequently learn a visual BC agent from it, removing the motion planner dependency. In step two, we learn a vision-based agent using an asymmetric actor-critic framework. This step is further expedited via proper weight initialization, BC trajectory-guided RL training, and entropy coefficient tuning, making our method highly sample-efficient. Our visual policy combined with domain randomization demonstrates successful zero-shot transfer to unseen environments with new visual domains and distractors. Beyond zero-shot transfer, fine-tuning the policy in the real world by learning a vision-based critic or applying simulation-to-real techniques [36] is our definite future work to realize simulation-to-real transfer.

## Acknowledgments

This research is supported by the Annenberg Fellowship from USC, NAVER AI Lab, and NSF NRI-2024768. We thank our colleagues from the CLVR lab and RESL for the valuable discussions that considerably assisted the research.

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
