# OpenReview forum: "Distilling Motion Planner Augmented Policies into Visual Control Policies for Robot Manipulation"
_robot-learning.org/CoRL/2021/Conference — CoRL2021 Poster_

### Official Review · Reviewer_acdh · 2021-07-22

**Originality:** Good
**Technical Quality:** Very Good
**Clarity Of Presentation:** Very Good
**Impact:** 3

**Recommendation:**

Weak Accept: I recommend accepting the paper, but will not argue for my recommendation if the majority of other reviewers have a different opinion.

**Summary:**

This paper proposes a technique for transferring a learned policy from privileged state information to images. The method can complete cluttered manipulation tasks after about 1 million training steps (a few days of real robot time), whereas appropriate baselines are unable to make any progress. This method combines two key ideas from prior work: the MoPA-RL framework which can solve cluttered manipulation tasks with privileged state information, and asymmetric actor-critic, which allows for efficient transfer to a visual policy.

**Issues:**

-Since the RRT is going to produce different paths for the same/similar start & goal based on the random seed, it seems like you could run into issues with BC based on this, especially if the policy network outputs a single Gaussian over the action vector. Does the method have a way to compensate for this?

-Why does BC cause smoothing, and seemingly do something smarter than averaging over actions (which could cause issues in some environments)? Why isn’t multimodality of actions an issue? Are there cases where it is? Answering some of these questions would strengthen this paper.

-I find the description of the MoPA vs direct action spaces as “different spaces” misleading. They have different domains, but they aren’t different spaces. A reader might think you’re adding additional dimensions to the action vector space, rather than expanding the domain of the action vector.

-I also do not entirely agree that the different action spaces “lead to different transition function … and reward function” (line 117). For the actions within the direct action subset, the transition and reward are the same, inputs and outputs shouldn’t change. In fact, that’s the reason why transferring the critic works, because the transition/reward (and hence Q-value) functions are unchanged. This is a minor point that could be clarified in the text.

Questions:

-For table 1, how many environment steps of training were used? I could not find that number. Perhaps put it in section 4.3. I also wonder, more generally, how much success in MoPA pretraining is needed? Does it still work if you stop pretraining before the task is “mostly solved”?

-The discussion of HER was not sufficient. I did not follow the explanation of why it’s not applicable. From a quick read of the HER paper, it seems like it should be applicable.

Minor comments:

-On line 261, consider removing “extremely”, I would not consider one million time steps extremely sample efficient. Perhaps replace with “far more sample-efficient than ...”

-Table 1: in the Sawyer Lift experiment, you bolded “Ours” but “Ours (w/o BC smoothing)“ had a better score, I think that should be bolded instead.

-Table 1&2: consider adding Up and Down arrows to indicate which metrics should be high and which should be low. That would save the reader from having to go back and re-read what the acronyms mean.

-Consider running more seeds, 3 seems like too small of a number to have much confidence in.

-There is a comma missing on line 61. I would have written it “in terms of success rate, sample efficiency, and average path length”

-Section A.3 is not particularly helpful because it’s hard to see visually which is smoother. Perhaps show only one trajectory, or split the graphs up more. You could also try connecting the points with lines. More importantly, what is being graphed? Is it end-effector position? That’s not explicitly state anywhere, I just guessed.


**Reviewer Expertise:**

Good: General knowledge of the area

**Strengths And Weaknesses:**

Strengths:

-An elegant combination of existing ideas

-Compelling results on difficult manipulation tasks

-Addresses an important problem -- efficiently learning a visuomotor policy for complex manipulation tasks

Weaknesses:

-No real robot experiment. Although the pandemic made this difficult, without such an experiment the suggestions/claims about sim2real transfer are not supported. There are many things other than textures, colors, and lighting which vary between simulations and the real world, such as physical properties, geometric differences, controller differences, etc. So the results for evaluating transfer are not convincing.

-It seems the actor still requires privileged state information, even during distillation. This is potentially a big limitation, because further training on a real robot is not possible without training a visual critic from scratch. To address this, perhaps one could train a visual critic during this whole process (using both step 1 and step 2 data, via simulation rendering + D.R.)? Then at the end you could conceivably fine-tune in the real world with efficiency boosts from all the pretraining.


**Summary Of Recommendation:**

With the addition of a real robot result quantifying success on a task using the proposed method, I would change to a strong accept. This paper combines existing methods elegantly and has strong results. However, a real robot experiment is missing, and I do not see any important insights or theoretical advances. There are a few components of the method which I believe limit its usefulness, which I discuss further in the Issues section.

---

> ### Author Response · Authors · 2021-08-30
> **Response to Reviewer acdh (1/2)**
>
> We appreciate your excellent suggestions and are excited to try them. Following are answers to your questions:
>
>
> &nbsp;
>
> **No real robot experiment. Although the pandemic made this difficult, without such an experiment the suggestions/claims about sim2real transfer are not supported.**
>
> We agree that showing sim2sim transfer is not sufficient to demonstrate the sim2real transfer capability. But, due to the current pandemic situation, we were unable to test our framework on a real robot setting. Our main motivation of this work is to distill state-based policies into visual policies that also function without requiring the motion planner. Our motivation and method is orthogonal to sim2real works where the main focus is on narrowing the domain gap between simulation and the real world. Although, we are excited to try this as a future work for our method by combining our approach with sim2real techniques such as [a] where the learnt policy is adapted to different target scenarios with varying visual as well as dynamics gaps without requiring state information, or by learning a visual critic which can be fine-tuned online.
>
>
> &nbsp;
>
> **The actor still requires privileged state information, even during distillation. … To address this, perhaps one could train a visual critic during this whole process (using both step 1 and step 2 data, via simulation rendering + D.R.)?**
>
> Thanks for the suggestion. The actor only takes the robot state and image as input, which can be directly used on a real robot. But during distillation, we need access to privileged information to train a critic.
>
> We agree that learning a visual critic does not require such privileged information, so this is a promising direction for realizing sim2real transfer by allowing fine-tuning the policy in the real world. As per the suggestion, we tried learning a visual critic yet we only obtained sub-optimal results for Sawyer Push and Sawyer Lift until 3M environment steps. Although we could not get successful visual critics, we believe this can be improved with recent advances in RL from pixels [b,c] and leave this as future work.
>
>
> &nbsp;
>
> **Does the method have a way to compensate for multi-modal paths generated by RRT?**
>
> We appreciate the point raised by the reviewer. As the reviewer mentioned, RRT can generate multiple routes for similar start and end goal positions. This could be a problem for certain tasks with multi-modal solutions. However, we did not observe this problem in our tasks. The paths generated from different motion planner calls in the MoPA-RL do not exhibit large variations. Although, in case of tasks, where this variation could be large and therefore problematic for BC to learn over multi-modal actions, it could be resolved by using an optimal planner like RRT* that converges to the shortest path.
>
>
> &nbsp;
>
> **Why does BC cause smoothing, and seemingly do something smarter than averaging over actions (which could cause issues in some environments)? Why isn’t multimodality of actions an issue? Are there cases where it is?**
>
> The main issue that we face with using the MoPA-RL expert trajectories directly is due to their jittery motion. This is usually the case in the presence of obstacles where the smoothness of trajectories is compromised in order to prioritize obstacle avoidance by the motion planner. Such jerky trajectories make it difficult for the RL agent to learn from them while interacting with the environment. Therefore, BC smoothing helps to overcome it. BC smoothing might cause issues during averaging in certain scenarios, but as mentioned in the above response, we do not observe such a problem in our tasks because the variability in the multiple Motion Planner trajectories for any given state is subtle.

---

> > ### Author Response · Authors · 2021-08-30
> > **Response to Reviewer acdh (2/2)**
> >
> > **I find the description of the MoPA vs direct action spaces as “different spaces” misleading.**
> >
> > Indeed, as Reviewer acdh mentioned, the direct action space is a subset of the augmented action space -- they have the same dimension but different interval bounds, which are larger for the augmented MDP with the motion planner actions. As the reviewer pointed out correctly, the reward and transition functions of both MDPs are the same, but we differentiate notations to specify their different domains. We clarify this point in the updated paper in L119-121, Section 3.1.
> >
> >
> > &nbsp;
> >
> > **For table 1, how many environment steps of training were used?**
> >
> > For the plots and other results in the paper, we train MoPA-RL for 1M environment steps. We further train our method and report our results after an additional 2M steps, with a total of 3M steps combined. For a fair comparison, we report results for all other RL baselines not requiring MoPA-RL after 3M environment steps as well.
> > To address the above comment, we update the information about the number of environment steps used for training in Table 1 caption and also update Figure 3 accordingly.
> >
> >
> > &nbsp;
> >
> > **Does it still work if you stop pre-training before the task is “mostly solved”?**
> >
> > To address reviewer's concern about if our method still works if we stop MoPA pre-training before the task is “mostly solved”, we use suboptimal checkpoints from MoPA-RL (69.2% success rate for Sawyer Push; 41.4% and 60.6% success rate for Sawyer Lift) to train our proposed framework. We find the proposed method to have a near 100% success rate using these suboptimal checkpoints. In short, the proposed method can still work very well when MoPA-RL cannot complete the task reliably. We discuss the results in detail in L429-439, Section A.4.
> >
> >
> > &nbsp;
> >
> > **The discussion of HER was not sufficient.**
> >
> > HER is designed to train a __goal-conditioned__ policy efficiently. However, our target tasks are not goal-conditioned, so HER cannot be directly used to solve these tasks.
> >
> > HER could be used to learn a goal-conditioned **low-level** controller (i.e., neural motion planner), but then the high-level policy needs to be distilled separately for explicitly specified sub-goals for the low-level controller. Instead, we distill both low-level and high-level policies into a single visual policy. We agree that the statement in the original submission might not be clear in itself. Thus, we clarified this justification in L90-94, Section 2.
> >
> >
> > &nbsp;
> >
> > **On line 261, consider removing “extremely”, perhaps replace it with “far more sample-efficient than ...”**
> >
> > We update the quoted line as per the reviewer’s suggestion.
> >
> >
> > &nbsp;
> >
> > **Table 1: in the Sawyer Lift experiment, you bolded “Ours” but “Ours (w/o BC smoothing)“ had a better score, I think that should be bolded instead.**
> >
> > Thank you for pointing it out. We updated Table 1 with the above mentioned correction regarding the bold values for best score.
> >
> >
> > &nbsp;
> >
> > **Table 1&2: consider adding Up and Down arrows to indicate which metrics should be high and which should be low**
> >
> > Thank you for the great suggestion! We updated both Table 1 and 2 accordingly.
> >
> >
> > &nbsp;
> >
> > **Consider running more seeds, 3 seems like too small of a number to have much confidence in.**
> >
> > As per request of Reviewer acdh, we ran one additional seed and found it consistent with the original results (see Table 1 and Figure 3 in the updated draft). We are running experiments with one more seed and will include results with *5 random seeds* in the revised version.
> >
> >
> > &nbsp;
> >
> > **Section A.3 is not particularly helpful because it’s hard to see visually which is smoother. More importantly, what is being graphed? Is it end-effector position?**
> >
> > Figure 8 in Section A.3 visualizes the _end-effector position_ of MoPA-RL and BC smoothed trajectories.
> >
> > We agree with the reviewer that the original visualization was difficult to see the difference. Following the reviewer's suggestion, we updated Figure 8, and the jittery motion of MoPA-RL trajectories is now more evident.
> >
> >
> > &nbsp;
> >
> > **There is a comma missing on line 61. I would have written it “in terms of success rate, sample efficiency, and average path length”**
> >
> > Thank you for spotting out the typos. We fixed them in the paper as suggested at L59, in the updated manuscript.
> >
> >
> > &nbsp;
> >
> > **References**
> >
> > [a] Zhang et al., Policy Transfer across Visual and Dynamics Domain Gaps via Iterative Grounding, RSS 2021.
> >
> > [b] Yarats et al., Mastering Visual Continuous Control: Improved Data-Augmented Reinforcement Learning, arXiv 2021
> >
> > [c] Laskin et al.,  Reinforcement Learning with Augmented Data, NeurIPS 2020

---

> > > ### Comment · Reviewer_acdh · 2021-09-02
> > > **Response from reviewer**
> > >
> > > Thank you for your reply. It's unfortunate that the visual critic did not produce good results. I am still hesitant about the privileged information requirement of this method, as this seriously undermines the applicability of the work. I will leave the rating as-is.

---

### Official Review · Reviewer_M4xK · 2021-07-23

**Originality:** Good
**Technical Quality:** Very Good
**Clarity Of Presentation:** Very Good
**Impact:** 3

**Recommendation:**

Weak Accept: I recommend accepting the paper, but will not argue for my recommendation if the majority of other reviewers have a different opinion.

**Summary:**

This paper proposes a learning approach for manipulation tasks in obstructed environments.
It has a hybrid aspect, where a sampling-based (thus model/state-based) motion planning algorithm is combined with a model-free reinforcement learning formulation.
As an extension of a prior work, here the focus is on obstructed environments, for which RL methods are prone to fail, whereas sampling-based motion planners are effective.
However, switching to a sampling-based motion planner during execution or learning is costly and also using them in the real-world is harder due to their dependency on full state knowledge.
With this motivation, a two-stage learning approach is proposed.
Initially a policy is learned based on the state-space representation using motion planning with RL.
Then, an asymmetric actor-critic framework is used to train a policy that is now observation-based, i.e. visual perception data.
The initially learned state-based policy is imitated by behavioral cloning to provide (smoothed) expert data for training the observation-based policy.
The results demonstrate the efficiency of the proposed framework on three manipulation tasks.

**Issues:**

My comment on domain randomization mentioned above (Strengths & Weaknesses) could be an improvement point.

Another recommendation is to include some limitation and discussion of the proposed approach. Such a critical evaluation of the overall architecture, where it fails, for which tasks it might be applicable (or not), what are lessons learned, etc. would be valuable for the community. Especially, for example the dependency of the success of the whole framework to the 'weight initialization' is highly interesting, which might be further elaborated.

**Reviewer Expertise:**

Very good: Comprehensive knowledge of the area

**Strengths And Weaknesses:**

The paper showcases the strengths of combining model-based methods with model-free approaches. In that regard, the motivation of the proposed approach is clearly explained w.r.t. the target scenarios, and the execution of the framework is technically sound and justified by several manipulation tasks.

The only slight weakness is the domain randomization experiments, as the distractors seem to be weak, and there isn't much analysis on the performance.

**Summary Of Recommendation:**

The paper proposes a novel approach combining state-based motion planners with model-free reinforcement learning methods for tackling challenging robot manipulation tasks. Even though it is an incremental update on prior work, the extension is well motivated and justified by demonstrating its performance on a range of scenarios. Overall, the paper is well structured and written. It can be further improved by a critical discussion on the limitations of the proposed approach and a deeper analysis on the domain randomization experiments.

---

> ### Author Response · Authors · 2021-08-30
> **Response to Reviewer M4xK**
>
> We appreciate the thorough reviews and insightful suggestions. Below we try to address them:
>
>
> &nbsp;
>
> **The only slight weakness is the domain randomization experiments, as the distractors seem to be weak, and there isn't much analysis on the performance.**
>
> To address the reviewer’s comments, we illustrate an additional unseen scenario with varying object sizes and texture in Section 4.4 and we ran experiments again on all three tasks for 1M environment steps. As we report in Table 2, our policy performs well across different unseen scenarios (shown in Figure 4), and obtains ~97-99% success rate consistently for all tasks, in a zero-shot transfer, i.e., without requiring fine-tuning.
>
> By distilling the motion planner into a visual policy, we lose the collision avoidance capabilities of the motion planner. If the environment changes drastically, then the policy might lead to unwanted collisions. We also add a more detailed analysis of key points for the success of our framework and potential limitations in L496-523, Section D.
>
>
> &nbsp;
>
> **Include some limitations and discussion of the proposed approach.**
>
> Our method mainly benefits from the following aspects:
>
> * Appropriate weight initialization for the actor and the critic:
> This step is crucial for efficiently distilling the state-based Motion Planner augmented policies into visual control policies. As reported in Section A.2, without initialization, the policy does not begin to learn any of the tasks even after 1M environment steps making it sample inefficient for such complex manipulation tasks in obstructed environments.
>
> * Using BC smoothing for MoPA-RL trajectories:
> Secondly, we observe that BC smoothing of the Motion-Planner trajectories is important for our method. Our explanation for this is that since the Motion-Planner-based trajectories are usually jittery, the RL agent finds it difficult to optimize over those expert trajectories.
>
> * Tuning the alpha parameter for controlling exploration-exploitation tradeoff:
> Another important aspect of our learning framework depends on choosing the appropriate value of alpha, the temperature coefficient in the SAC training which controls exploration vs exploitation tradeoff. Since we utilize prior knowledge obtained by the state agent, we use alpha smaller than the final alpha value of MoPA-RL. This helps us to accelerate the distillation step.
>
> Limitations: By distilling the motion planner into a visual policy, we lose the collision avoidance capabilities of the motion planner. If the environment changes drastically, then the policy might lead to unwanted collisions.
>
> Based on the comment above, we agree that a more profound discussion section highlighting the key points as well as limitations can be useful for the readers. Therefore, we added L496-523, Section D as a discussion section in the updated paper.

---

> > ### Comment · Reviewer_M4xK · 2021-09-01
> > **Response to rebuttal**
> >
> > Thank you for the clarifications and extending your discussion on limitations of this proposed approach. I'm going to keep my recommendation at a solid 'weak-accept'.

---

### Official Review · Reviewer_rTHv · 2021-07-28

**Originality:** Good
**Technical Quality:** Very Good
**Clarity Of Presentation:** Excellent
**Impact:** 3

**Recommendation:**

Weak Accept: I recommend accepting the paper, but will not argue for my recommendation if the majority of other reviewers have a different opinion.

**Summary:**

This paper presents an approach for distilling motion-planner augmented policies that operate in state space into visual control policies that operate without augmented actions from a motion planner. The first stage of this approach is to use Motion Planner Augmented Reinforcement Learning (MoPa-RL) to train an actor and critic model that operates in state space and has augmented actions from a motion planner to learn a policy to complete a task. The second stage of this approach uses behavioral cloning to train an actor network that operates in visual space and without a motion planner by using demonstrations from the expert MoPa-RL agent. The third stage trains an asymmetric actor-critic model to learn a visual motor policy and state-based critic that do not use the augmented action space by using demonstrations from the behavioral cloning agent and weight initializations from the MoPa-RL expert agent. The authors conduct experiments in 3 object manipulation domains (object pushing, object lifting, and object assembly) that involve obstacles and visual obstructions, and compare their full method to ablated versions of their method as well as other state-of-the-art RL and learning from demonstration approaches. This paper also demonstrates the robustness of their approach to domain randomization.

**Issues:**

The authors need to explicitly validate the sample complexity for each of the stages of the proposed approach, and they also need to include the experimental details mentioned in the "Weaknesses" previously mentioned. If these are addressed, then this paper stands to present a good contribution and deserve publication.

**Reviewer Expertise:**

Very good: Comprehensive knowledge of the area

**Strengths And Weaknesses:**

Strengths:
- This paper presents a novel and interesting approach for distilling a motion-planner augmented policy that operates in state space into a visual control policy that does not rely on motion planning at all. The method is also extremely clear and well-presented, and the reviewer commends the authors for their clarity. Overall, this method is a very good contribution for enabling motion planners to be useful for learning visual motor policies without requiring a motion planner or hand-crafted state space to be used at inference time. Overall this paper and method deserve praise and could provide a good contribution to the field, and the reviewer really enjoyed reading the paper.
- This paper includes various ablations of their method and plenty of comparisons to other state-of-the-art approaches for this robot policy learning task, and report quantitative results on average success rate, episode length, and the discounted returns.
-The figures in the paper are clear and useful for understanding the method, robot domains, and results.

Weaknesses:
- This proposed approached requires several stages of environment interactions that are unaccounted for the in quantitative results, which questions how sample efficient this method really is. The first stage requires MoPA-RL to interact and learn from the environment. The second stages requires behavioral cloning to occur (to get a smoother policy), and then they also collect more samples using the behaviorally cloned agent before doing the final stage of learning with the asymmetric actor, which further requires environment interactions. However, it appears none of the results reported in this paper appear to include how long these previous stages up to the last one take, or at least it is not clear from the graphs at which part each stage occurs. The authors should be explicit in their graphs about how many samples each of the stages require if they wish to make claims that their overall method is more sample efficient than other approaches. This is especially important since the authors emphasize how important the BC step is for good results (re: Sawyer assembly task)
- The authors claim in the related work that "However, using HER is not applicable for composite tasks involving goals associated with multiple sequential steps", but this statement needs to be justified since it is not clear why this would be true, since rewards can just be relabeled based on whether they achieve a subgoal (defined by composite task) or not.
- The authors should clarify what the state space was used in the experiments, and they should also clarify the size of the images used for the observations.
- In section 3.3.1, the authors claim to store transitions from MoPA-RL into a demonstration dataset for later stages. Does this include transitions that include motion-planner augmented actions?
- The authors claim they conduct their experiments in a sparse reward setting, but in the the tasks they use a dense reward around an epsilon ball of the goal proportional to the distance, they should specify how large epsilon is.
- The authors do not specify what the gamma value is, which is important for the sparse reward setting because if it is not less than 1, then there is no useful distinction between the ASR and ADR metric.
- The ASR metric for MoPA-RL against the proposed method across all three tasks is extremely similar (at max, a ~5% improvement in the lift task, and only a ~1% improvement in the other two tasks). Although AEL and ADR are improved, the ASR is the most interesting metric when considering whether these methods can operate in tasks with obstacles where vision is important for completing it. This makes the results much less convincing: the authors should choose a task where MoPA-RL is not able to reliably complete the task to demonstrate the importance of this distilling approach (although the reviewer recognizes it is useful regardless of success rate since it removes motion planning from the loop).

**Summary Of Recommendation:**

This paper presents a novel and interesting approach for distilling motion-planner augmented policies into visual motor policies, and is very easy to read and is overall enjoyable. The reviewer really enjoys this paper and the contribution it offers, but the the experimental validation of the sample and computational efficient of this method due to the MoPA-RL expert and BC stages is not included and brings into question the efficacy of this method. There are also other important experimental details that should be included. Overall this paper has the potential to be a good contribution to the field (and is very accessible and easy to read), but requires further experimental validation for each of the stages and needs other experimental details to be included before publication.

---

> ### Author Response · Authors · 2021-08-30
> **Response to Reviewer rTHv**
>
> We appreciate your encouraging reviews and suggestions. Here is our response:
>
>
> &nbsp;
>
> **The authors should be explicit in their graphs about how many samples each of the stages require.**
>
> Thank you for your suggestion on graph rendering! For an easier comparison of sample efficiency across different methods, we updated Figure 3, which now shows the number of samples used by each stage.
>
> However, the BC step is still not included in the graph as BC is trained __fully offline__ (i.e., it does not require any environment interactions). Instead, we made it clear how many epochs (and wall-clock time) we train the BC policy in Table 8 and mentioned it in the caption of Figure 3.
>
>
> &nbsp;
>
> **"However, using HER is not applicable for composite tasks involving goals associated with multiple sequential steps" needs to be justified.**
>
> HER is designed to train a __goal-conditioned__ policy efficiently. However, our target tasks are not goal-conditioned, so HER cannot be directly used to solve these tasks.
>
> As Reviewer rTHv mentioned, HER could be used to learn a goal-conditioned **low-level** controller (i.e., neural motion planner), but then the high-level policy needs to be distilled separately for explicitly specified sub-goals for the low-level controller. Instead, we distill both low-level and high-level policies into a single visual policy. We agree that the statement in the original submission might not be clear in itself. Thus, we clarified this justification in L90-94, Section 2.
>
>
> &nbsp;
>
> **Clarify what state and observation spaces were used in the experiments.**
>
> Thank you for spotting the missing information about observation space! For all the image-based methods and environments, we use an image of size 32x32 and robot state (7-dimensional joint angles) as an observation. We specified it in L215-216, Section 4.1.
>
> The state space for each environment is the same as MoPA-RL [1], and the details can be found in Section C, appendix.
>
>
> &nbsp;
>
> **In Section 3.3.1, the authors claim to store transitions from MoPA-RL into a demonstration dataset for later stages. Does this include transitions that include motion-planner augmented actions?**
>
> The transitions from MoPA-RL are collected in _low-level action space_, not the motion-planner augmented actions. We made this clearer in L164-165.
>
>
> &nbsp;
>
> **How large is the epsilon for reward functions?**
>
> We named our task _a sparse reward setting_ following [a], which only gets rewards in an epsilon-region near the goal. Specifically, we set $\epsilon$ to 0.1 and 0.3 for Sawyer Push and Sawyer Assembly respectively, while the distance between the initial and goal state is around 1.2. We clarified this in the updated manuscript, L227-230.
>
>
> &nbsp;
>
> **The ASR metric for MoPA-RL against the proposed method across all three tasks is extremely similar.**
>
> As Reviewer rTHv pointed out, the goal of our method is to get rid of access to state information as well as motion planning; thus, outperforming MoPA-RL in ASR is not our goal. Nonetheless, our method shows comparable performance with MoPA-RL and enables real-world transfer of the learned policy. Moreover, our method reduced AEL approximately 2 times with fine-tuning.
>
> To address the reviewer's concern about how the proposed method performs when MoPA-RL is not able to complete the task reliably, we use suboptimal checkpoints from MoPA-RL (69.2% success rate for Sawyer Push; 41.4% and 60.6% success rate for Sawyer Lift) to train our proposed framework. Surprisingly, we find the proposed method to have a near 100% success rate using these suboptimal checkpoints. In short, the proposed method can still work very well when MoPA-RL cannot complete the task reliably by fine-tuning from the smoothed behaviors. We discuss the results in detail in L429-439, Section A.4.
>
>
> &nbsp;
>
> **What is the gamma value?**
>
> For all our experiments, we used $\gamma=0.99$. We specified this information in L258, Section 4.3.
>
>
> &nbsp;
>
> **References**
>
> [a] Riedmiller et al., Learning by Playing - Solving Sparse Reward Tasks from Scratch, ICML 2018.

---

> > ### Comment · Reviewer_rTHv · 2021-09-03
> > **Post-response**
> >
> > I thank the reviewers for their clarifications on these raised points, it has helped improve my understanding of the paper. I will still be keeping my recommendation at a strong "weak accept".

---

### Meta-Review · Area_Chair_Z7bx · 2021-08-13

**Recommendation:** Accept (Poster)
**Confidence:** 5

**Metareview:**

All three reviewers were generally positive about this paper. They agree that the method is novel and interesting, albeit somewhat incremental and an "elegant combination of existing ideas”. Reviewers comment that the paper is particularly well written. However, reviewers do raise some important points that should be addressed in a rebuttal. For example, further details on the sample and computational efficiency are important for a full evaluation, and some insights into the limitations and failure cases would be helpful. I would like to see the authors address the comment from Reviewer acdh about whether the actor still requires privileged state information during distillation, and how this relates to implementation on a real robot. Reviewers comment on the lack of real-world experiments, particularly given that sim-to-real transfer features prominently in this work, which is an important limitation. However, I do not necessarily expect authors to provide real-world results for the paper to be accepted. Overall, this is a good paper but I encourage authors to address the reviewers' criticisms and major queries in a rebuttal, in order to maintain good reviews.

------

Following the reviews, the authors have addressed a number of issues raised by the reviewers, and new experiments with a visual critic have been provided. There is still concern about the applicability of the method to real-world tasks, given the lack of real-world experiments and the sub-optimal results for the visual critic. However, the reviewers still found the overall idea appealing and with potential for interesting future work, and so all three reviewers recommended to accept the paper.

---

> ### Author Response · Authors · 2021-08-30
> **Response to Area Chair Z7bx**
>
> Thank you for summarizing the most important points from the reviewers. We first address the common concerns here and individually answer each reviewer’s comments. According to the reviewers’ feedback, we made changes in our paper and marked them in red.
>
>
> &nbsp;
>
> **Further details on the sample and computational efficiency.**
>
> We updated the graphs in Figure 3 to also represent samples used by the individual stages of our approach. Now, it should be easier to compare the sample efficiency across different methods.
>
> The BC step is not included in the graph because it is trained fully offline (i.e., it does not require any environment interactions). Instead, we made it clear how many epochs (and wall-clock time) we train the BC policy in Table 8 and the caption of Figure 3.
>
>
> &nbsp;
>
> **The actor still requires privileged state information during distillation, and how this relates to implementation on a real robot.**
>
> The actor only takes the robot state and image as input, which can be directly used on a real robot. But during distillation, we need access to privileged information to train a critic.
>
> As Reviewer acdh mentioned, learning a visual critic does not require such privileged information, so this is a promising direction for realizing sim2real transfer by allowing fine-tuning the policy in the real world. As per the suggestion, we tried learning a visual critic yet we only obtained sub-optimal results for Sawyer Push and Sawyer Lift until 3M environment steps. Although we could not get successful visual critics, we believe this can be improved with recent advances in RL from pixels [a,b] and leave this as future work.
>
>
> &nbsp;
>
> **Some insights into the limitations and failure cases.**
>
> By distilling the motion planner into a visual policy, we lose the collision avoidance capabilities of the motion planner. If the environment changes drastically, then the policy might lead to unwanted collisions.
>
> We also discuss key insights and potential limitations of our method in more detail in Section D.
>
>
> &nbsp;
>
> **The lack of real-world experiments. However, I do not necessarily expect authors to provide real-world results for the paper to be accepted.**
>
> We agree that showing sim2sim transfer is not sufficient to demonstrate the sim2real transfer capability. But, due to the current pandemic situation, we were unable to test our framework on a real robot setting. Our main motivation of this work is to distill state-based policies into visual policies that also function without requiring the motion planner. Our motivation and method is orthogonal to sim2real works where the main focus is on narrowing the domain gap between simulation and real world. Although, we are excited to try this as a future work for our method by combining our approach with sim2real techniques such as [c] where the learnt policy is adapted to different target scenarios with varying visual and dynamics gaps without requiring state information.
>
>
> &nbsp;
>
> **References**
>
> [a] Yarats et al., Mastering Visual Continuous Control: Improved Data-Augmented Reinforcement Learning, arXiv 2021
>
> [b] Laskin et al.,  Reinforcement Learning with Augmented Data, NeurIPS 2020
>
> [c] Zhang et al., Policy Transfer across Visual and Dynamics Domain Gaps via Iterative Grounding, RSS 2021.

---

### Decision · Program_Chairs · 2021-09-13

**Decision:**

Accept (Poster)

**Comment:**

All three reviewers were generally positive about this paper. They agree that the method is novel and interesting, albeit somewhat incremental and an "elegant combination of existing ideas”. Reviewers comment that the paper is particularly well written. However, reviewers do raise some important points that should be addressed in a rebuttal. For example, further details on the sample and computational efficiency are important for a full evaluation, and some insights into the limitations and failure cases would be helpful. I would like to see the authors address the comment from Reviewer acdh about whether the actor still requires privileged state information during distillation, and how this relates to implementation on a real robot. Reviewers comment on the lack of real-world experiments, particularly given that sim-to-real transfer features prominently in this work, which is an important limitation. However, I do not necessarily expect authors to provide real-world results for the paper to be accepted. Overall, this is a good paper but I encourage authors to address the reviewers' criticisms and major queries in a rebuttal, in order to maintain good reviews.

------

Following the reviews, the authors have addressed a number of issues raised by the reviewers, and new experiments with a visual critic have been provided. There is still concern about the applicability of the method to real-world tasks, given the lack of real-world experiments and the sub-optimal results for the visual critic. However, the reviewers still found the overall idea appealing and with potential for interesting future work, and so all three reviewers recommended to accept the paper.